# Semantic Modelling Approach for Safety-Related Traffic Information Using DATEX II

J. Javier Samper-Zapater [1],*, Julián Gutiérrez-Moret [1], Jose Macario Rocha [1], Juan José Martinez-Durá [1] and Vicente R. Tomás [2]

1   IRTIC—Research Institute on Robotics and Information and Communication Technologies, University de València, 46980 Valencia, Spain; jgutierrez@irtic.uv.es (J.G.-M.); Jose.M.Siqueira@uv.es (J.M.R.); juanjo@irtic.uv.es (J.J.M.-D.)

2   Department of Computer Science and Engineering, University Jaume I, 12061 Castellón de la Plana, Spain; vtomas@uji.es

\*   Correspondence: jsamper@irtic.uv.es

**Abstract:** The significance of Linked Open Data datasets for traffic information extends beyond just including open traffic data. It incorporates links to other relevant thematic datasets available on the web. This enables federated queries across different data platforms from various countries and sectors, such as transport, geospatial, environmental, weather, and more. Businesses, researchers, national operators, administrators, and citizens at large can benefit from having dynamic traffic open data connected to heterogeneous datasets across Member States. This paper focuses on the development of a semantic model that enhances the basic service to access open traffic data through a LOD-enhanced Traffic Information System in alignment with the ITS Directive (2010/40/EU). The objective is not limited to just viewing or downloading data but also to improve the extraction of meaningful information and enable other types of services that are only achievable through LOD. By structuring the information using the RDF format meant for machines and employing SPARQL for querying, LOD allows for comprehensive and unified access to all datasets. Considering that the European standard DATEX II is widely used in many priority areas and services mentioned in the ITS Directive, LOD DATEX II was developed as a complementary approach to DATEX II XML. This facilitates the accessibility and comprehensibility of European traffic data and services. As part of this development, an ontological model called *dtx_srti*, based on the DATEX II Ontology, was created to support these efforts.

**Keywords:** safety road traffic; semantic web; LOD

## 1. Introduction

Domain ontologies express conceptualizations specific to a particular domain [1]. In transportation, many solutions have emerged. For example, in [2] a vehicle accident ontology has been defined, in [3] an ontology for traffic management and control has been defined and tested in a multiagent system. In [4], an implementation of a semantic web service discovery system for road traffic information has been developed. Details of existing traffic ontologies are analyzed in [5].

In distributed environments, such as in transportation, data are the key element for developing Intelligent Transport Systems (ITS). The specification of DATEX II XML/UML [6] provides a description of concepts and data structures pertaining to traffic. However, it is important to note that this description is primarily focused on the syntactic aspects and does not incorporate semantic meaning. Therefore, there is a necessity for a semantic model that describes the contents of DATEX II elements to facilitate linking these data with other vocabularies and ontologies available on the internet. Linked Open Data (LOD) emerges as a methodology for publishing and interlinking structured data on the web, following the principles of the Semantic Web. LOD offers a standardized framework for sharing and

connecting data from diverse domains and sources, enabling data integration, enrichment, and reuse. The utility of LOD can be described through various aspects, including data integration and enrichment, data discovery, data reuse, interoperability, and data quality.

Its usefulness has been demonstrated in a variety of domains, such as healthcare [7,8] or education [9,10].

In the transportation domain, the use of non-LOD formats produces some interoperability problems. For example, in the European project *CROCODILE corridor* [11], to make possible the exchange of information between operators, road administrators and end-users, a middleware was created to provide the translation of different *situation_publication*. A *situation_publication* is an identifiable instance of a traffic/travel situation comprising one or more traffic/travel circumstances that are linked by one or more causal relationships. Types differ between the different national implementations of DATEX II. However, the middelware requires upgrading when new data concepts are added to any DATEX implementation, making this solution neither efficient nor viable. So, we consider that the use of LOD would be a much more efficient solution, given that sharing the same semantic model (Ontology for DATEX II) could avoid the interoperability problems, since it avoids the semantic ambiguity of specific terms.

In the CEF Action LOD-RoadTran18 reference Nº 2018-EU-IA-0088 [12], our primary aim was to facilitate the effective reuse of real-time road traffic data in the Czech Republic and Spain. To achieve this objective, we developed the DATEX II Safety Road Traffic Information (SRTI) ontology called *dtx_srti*. This ontology, built on the foundation of DATEX II, underwent rigorous testing on diverse datasets using SPARQL queries at a small scale [13]. In addition, we invested considerable effort in creating tools for model conversion, adaptation, and validation, with specific attention given to the geospatial characteristics of the traffic data.

In order to accomplish our goals, we thoroughly analyzed existing DATEX II models, LOD general concepts, as well as relevant vocabularies and models related to traffic information. This paper provides comprehensive details on the development of the semantic model. The model features a primary ontology housing the fundamental concepts of the DATEX II standard, complemented by two secondary ontologies—one designed for road concepts and the other for administrative units. Considering that traffic events occur across national road networks, the project tasks include analyzing vocabularies, ontologies, and datasets likely to become sources of information linked to the new general ontological model of DATEX II SRTI. Another aspect to consider in various instances of the SRTI general model is territorial organization. Incidents occur in sections of specific roads belonging to municipalities, provinces, and autonomous communities. Therefore, similar to the analysis of roads, a comparable examination is undertaken in the project regarding administrative units or areas.

The separation became essential to manage diverse data domains effectively. The modules will facilitate scalability and will be interconnected with the main ontology. Once the different sub-ontologies are constructed, they merge into a main ontology, consolidating all the main concepts. This ontology only references the key concepts defined in the rest of the sub-ontologies.

Also, modularity is a crucial aspect to consider, as maintaining independent ontologies allows each one to be modified separately. Even within a single ontology, it is highly beneficial to adjust hierarchies based on structures, roles, etc. This approach enables the incorporation of various classifications within a single ontology, creating sub-taxonomies that can be modified and managed independently. In fact, due to the disparities in road and administrative unit concepts between the Czech Republic and Spain (countries participating in the project), we worked independently on the development of secondary ontologies, successfully integrating them in the final stages.

We delve into the most pertinent aspects, providing elaborate explanations of the decisions made regarding the current and potential future usage of vocabularies and datasets. The paper covers the definition of concepts, their inter-relationships, and the

utilization of individual instances instead of the generic and specific data types found in the UML/XML schema of DATEX II version 3.2 [6].

The purpose of this paper is to provide a detailed description of the core *dtx_srti* ontology, offering a semantic description of its contents based on the UML data model developed as part of the CEF Action.

The main contributions of this research are summarized as follows:

- A groundbreaking semantic modeling approach for traffic information was introduced through the development of a novel ontology known as *dtx_srti* [14]. This ontology, accompanied by a couple of secondary ontologies, serves as a comprehensive semantic vocabulary specifically designed to represent the SRTI DATEX II profile [15] in accordance with the Commission Delegated Regulation (EU) [16]. The primary purpose of this vocabulary is to facilitate seamless mapping between DATEX II version 3.2 and Linked Open Data (LOD) formats, enabling efficient interoperability and data exchange. By leveraging these semantic resources, a significant step forward has been taken in advancing the representation and integration of traffic information within the context of dynamic data utilization.
- Regarding the implementation aspect, the ontologies put forward in this study were constructed using RDF/OWL (Resource Description Framework/Web Ontology Language) [17]. In order to enhance the depth and breadth of knowledge representation, these ontologies have been interconnected with relevant external ontologies. To consolidate data in the RDF standard [18], a set of mapping functions has been devised, enabling automatic storage in a shared RDF repository and Endpoint service. Building upon this foundation, an array of sophisticated SPARQL queries has been formulated, designed as an API service. Furthermore, to foster widespread adoption within the research community, a user interface has been developed, streamlining the utilization of these resources. This user-friendly interface aims to facilitate seamless exploration and interaction with the ontologies, empowering researchers to harness their full potential.
- To ensure the robustness and accuracy of the developed ontologies, a comprehensive validation process was undertaken. Multiple tests were conducted, involving various queries on traffic incident data, administrative units, and road information. Furthermore, federated queries were designed to establish connections between the data generated by our LOD Converter and other SPARQL Endpoints, including the esteemed Spanish National Geographic Institute (IGN) [19]. These tests not only explored the potential of leveraging linked data from diverse datasets but also assessed the intricacies and challenges associated with executing complex SPARQL queries. In doing so, the tests aimed to identify and rectify common errors and shortcomings, contributing to the refinement and optimization of the overall system [20].

This paper is organized as follows: Section 2 highlights the required legal and technical aspects related to the research. Section 3 describes the semantic approach, focusing on the OWL Ontology. The definition and implementation of the Linked Open Traffic Data Model is described in Section 4. The procedure to validate this approach is described in Section 5. Finally, Section 6 concludes with key remarks and future works. All this content not only provides a picture of semantic modelling within Safety Traffic Data but also an example of research and implementation that can be partially replicated for the creation and analysis of other ontologies.

## 2. Exploration of Legal and Technical Framework

The initial phase of the research involved a thorough examination of the regulatory landscape encompassing the Public Sector Information (PSI) and Intelligent Transport Systems (ITS) Directives, both at the European and national levels. The primary objective was to identify the key legal and procedural considerations that should guide the development of a semantic model and subsequent architecture. To accomplish this, the analysis was divided into two distinct tasks: (1) scrutinizing legal and procedural aspects associated with the PSI Directive, which governs the access and reuse of public sector information and

(2) evaluating legal and procedural aspects relevant to the ITS Directive, which outlines the framework for intelligent transport systems.

By delving into these areas, the research sought to establish a comprehensive understanding of the regulatory landscape, providing essential insights to inform the subsequent development of a robust semantic model and architecture.

### 2.1. Analysis of the Public Sector Information Directive

The primary objective of this task was to conduct a comprehensive review of the existing legislation pertaining to PSI at both the European and national levels. This analysis encompassed key documents and related materials, including:

- The Open Data Directive ( (EU) 2019/1024), as well as the previous PSI Directives 2003/98/EC and its later revision 2013/37/EU.
- Commission Notice (CN) 2014/C 240/01, titled "Guidelines on recommended standard licenses, datasets, and charging for the reuse of documents".
- Directive 2007/2/EC, which pertains to the establishment of an Infrastructure for Spatial Information in Europe (INSPIRE).

For example, according to CN 2014/C 240/01 recommendations, to enhance the usability of public sector data and significantly elevate its value for subsequent reuse, datasets should adhere to the following criteria:

(a) Timely Release: publish datasets online promptly and in their original, unmodified form to ensure timely accessibility.
(b) Granularity and Completeness: publish and update datasets at the highest possible level of granularity to guarantee completeness.
(c) Stable Location: ensure datasets are published and maintained at a stable location, preferably at the highest organizational level within the administration, for easy access and long-term availability.
(d) Machine-Readable Formats: publish datasets in machine-readable and open formats (CSV, JSON, XML, RDF, etc.) to enhance accessibility.
(e) Rich Metadata Descriptions: describe datasets using rich metadata formats and classify them according to standard vocabularies (DCAT, EUROVOC, ADMS, etc.) to facilitate searching and interoperability.
(f) Accessibility through Dumps and APIs: make datasets accessible as data dumps (massive data outputs) and through application programming interfaces (APIs) to facilitate automatic processing.
(g) Explanatory Documents: accompany datasets with explanatory documents detailing the metadata and controlled vocabularies used, promoting the interoperability of databases.
(h) Feedback Mechanism: subject datasets to regular feedback from re-users through channels such as public consultations, comments boxes, blogs, automated reporting, etc., to maintain quality over time and encourage public involvement.

Additionally, the analysis involved an examination of open data initiatives, such as the APORTA Initiative of the Spanish Government [21], action plans, national strategies in each country, and the initial assessment of compliance levels.

### 2.2. Analysis of the Intelligent Transport Systems Directive

This task involved a comprehensive analysis of the norms outlined in the ITS Directive, specifically focusing on the publication of traffic data. The analysis encompassed the types of data to be published, the manner in which such data should be made available, and the implementation of the National Access Point (NAP) for road traffic information to meet the specified requirements.

A review of the state of the art was conducted, taking into consideration the ITS Directive 2010/40/EU [22], subsequent delegated regulations, associated legislation, and relevant standards. Notably, insightful information from analytical documents of the EU

EIP project [23], which focused on European harmonization within the ITS Directive, was also taken into account.

The analysis concluded that the ITS Directive and subsequent delegated regulations provide the necessary specifications to ensure compatibility, interoperability, and continuity in the implementation and operational use of data and procedures. These specifications aim to facilitate the following types of information or priorities:

1. Provision of EU-wide multimodal travel information services.
2. Provision of EU-wide Real-Time Traffic Information (RTTI) services.
3. Data and procedures for the provision of free Safety-Related minimum Traffic Information (SRTI).
4. Harmonized provision for an Interoperable EU-wide eCall.
5. Provision of information services for safe and secure parking places for trucks and commercial vehicles.
6. Provision of reservation services for safe and secure parking places for trucks and commercial vehicles.

It is crucial to emphasize that the required data collection for the SRTI service , initial scope of the project, encompasses the following events or conditions falling under at least one of the specified categories:

(a) Temporary slippery road;
(b) Presence of animals, pedestrians, obstacles, or debris on the road;
(c) Unprotected accident area;
(d) Short-term road works;
(e) Reduced visibility;
(f) Wrong-way driver;
(g) Unmanaged blockage of a road;
(h) Exceptional weather conditions.

Furthermore, the necessary information to be provided includes:

- Location of the event or condition;
- Category of the event or condition along with a brief description;
- Advice on driving behavior.

Summarizing, in accordance with the recommendations regarding the PSI and the specifications of the ITS directives, and after analyzing various related initiatives, we will be well-positioned to create and effectively utilize our semantic model. Next, we will discuss some aspects related to the use of controlled vocabularies, as recommended by various standards and specifications.

**3. Semantic Issues Related to Controlled Vocabularies**

A comprehensive analysis was conducted to explore semantic aspects, focusing on metadata approaches within the contexts of Intelligent Transport Systems (ITS) and Open Data. This research aimed to investigate the rules and standards implemented in different countries. For instance, the Interoperability Technical Standard (NTI-RISP) [24] was thoroughly examined in Spain. This standard aims to establish common conditions for the selection, identification, description, format, use, and provision of public sector documents and information resources. It forms a part of the Spanish National Interoperability Scheme [25].

NTI-RISP utilizes its own controlled vocabulary and defines specific metadata elements, some of which are mandatory. It also establishes various taxonomies for specific metadata. For example, it adopts the W3C Time Ontology [26] to specify time-related property values and incorporates standards such as ISO-8601 (date-time) and RFC4646 (tags for identifying languages). Furthermore, NTI-RISP outlines the acceptable values for the "Geographic coverage" property, which involve the use of identifiers related to geographic resources within the Spanish territory at the level of autonomous communities, cities, and provinces [27].

Additionally, in the case of DCAT-AP [28], the URIs are defined based on the Languages Name Authority List [29] provided by the Publications Office of the EU. Furthermore, the Catalogue of Single Point Access Coordinated Metadata [30], recommended by the EIP project [23] for the implementation or enhancement of any National Access Point (NAP), suggests the use of NUTS 0-3 [31] levels (Nomenclature of Territorial Units for Statistics) for the "area covered by publication".

During the analysis, it was also highlighted that NTI-RISP includes 22 primary sectors (listed in Annex IV of the regulation) [32], with one of them being the "Transporte" (Transport) concept that encompasses communication and traffic-related issues.

## 4. Design and Implementation of a Comprehensive LOD Model for Road Traffic

Upon establishing the initial scope of the project, a thorough analysis of the DATEX II model was conducted, with a primary focus on the Safety-Related Traffic Information (SRTI) profile [15]. This profile represents a subset of the complete DATEX II model (version 3.2). However, after extensive discussions, it was decided to extend the model beyond the SRTI profile by introducing new terms and relationships. This expansion allows the model to accommodate various types of information beyond SRTI, thus paving the way for future extensions. Additionally, previous semantic models related to traffic, such as [33–35], were taken into consideration.

In the development of the project, an ontology is constructed to define concepts related to traffic, its situations, and elements present in the DATEX II standard. The Protégé editor [36] tool, focusing on the OWL-DL ontology description language, is employed for this ontology's creation.

The concept of "situation" in DATEX II encompasses any traffic situation or event that could occur in a real-world scenario. While a detailed discussion of the situation element is beyond this paper's scope, a brief overview is necessary. DATEX II defines a hierarchy of situation types: those resulting from operator actions on the road, events unrelated to the road, or traffic-related situation elements. Within the latter, further subdivisions include Accidents, Abnormal Traffic... In addition to this hierarchy, each situation contains information about the event itself: its impact, duration, cause, and more. The hierarchical structure of the "SituationRecord" concept in the created ontology is depicted in Figure 1.

Existing methodologies and practical development experiences as METHONTOLOGY [37] share certain common steps, such as initiating construction by identifying the purpose and scope of the ontology and the knowledge acquisition needs for a specific domain. However, they differ in their approach and the subsequent steps undertaken.

For developing our model, we can distinguish eight clearly defined phases:

1. Purpose and Scope Definition: ensuring alignment with the intended purpose and clearly defining the scope.
2. Standardization and Modularity: establishing standards and promoting modularity by creating subdomains.
3. Ontology Reuse: leveraging existing ontologies for effective reuse of established concepts.
4. Basic Translation: initial translation of concepts into a formal ontology structure.
5. Refinement: iterative refinement of the ontology, considering structures, relationships, and roles.
6. Knowledge Extension: expanding knowledge by adding instances and specific examples.
7. Testing or Evaluation: conducting thorough testing and evaluation to ensure ontology effectiveness.
8. Documentation: comprehensive documentation of the ontology, including its structure and intended use.

More specifically, the construction of the model consists of three steps:

1. Knowledge Acquisition: Identifying basic classes or terms and their properties.
2. Definition: Identifying relationships between classes.

3. Specification of Constraints: Identifying constraints that will limit how descriptions can be formed.

This process is iterative, so once definitions are specified, successive refinements result in much more elaborate definitions. The use of reasoners like "Pellet" is crucial in organizing the model and verifying its consistency.

Information and knowledge related to traffic involve specific concepts such as the current state of the road network, regulations, restrictions, routes, meteorology, etc. These concepts, their properties, the relationships between them, and the specific terms used to designate them are crucial as they provide a basic infrastructure to organize and connect information elements in any specification.

Developing the conceptual infrastructure that we need would be a monumental task if we had to start from scratch. However, there have been significant initiatives in recent years aimed at easing the development process. Libraries of ontologies have been created as part of these initiatives, providing readily available resources that prove invaluable in constructing new ontologies. The knowledge already acquired, conceptualized, and expressed in a formal language within these ontology libraries can be reused to build a new ontology, allowing for the removal, addition, and/or modification of concepts as needed. It is important to highlight the reuse of pre-defined vocabularies such as GeoSPARQL, W3C TIME, QUDT, and others.

The core ontology is structured between two major groups: the representation of various traffic incidents (Situation Records) occurring within road networks (refer to Figure 1), and the geographical location aspects (GML, TPEG, OpenLR, and AlertC) associated with these incidents.

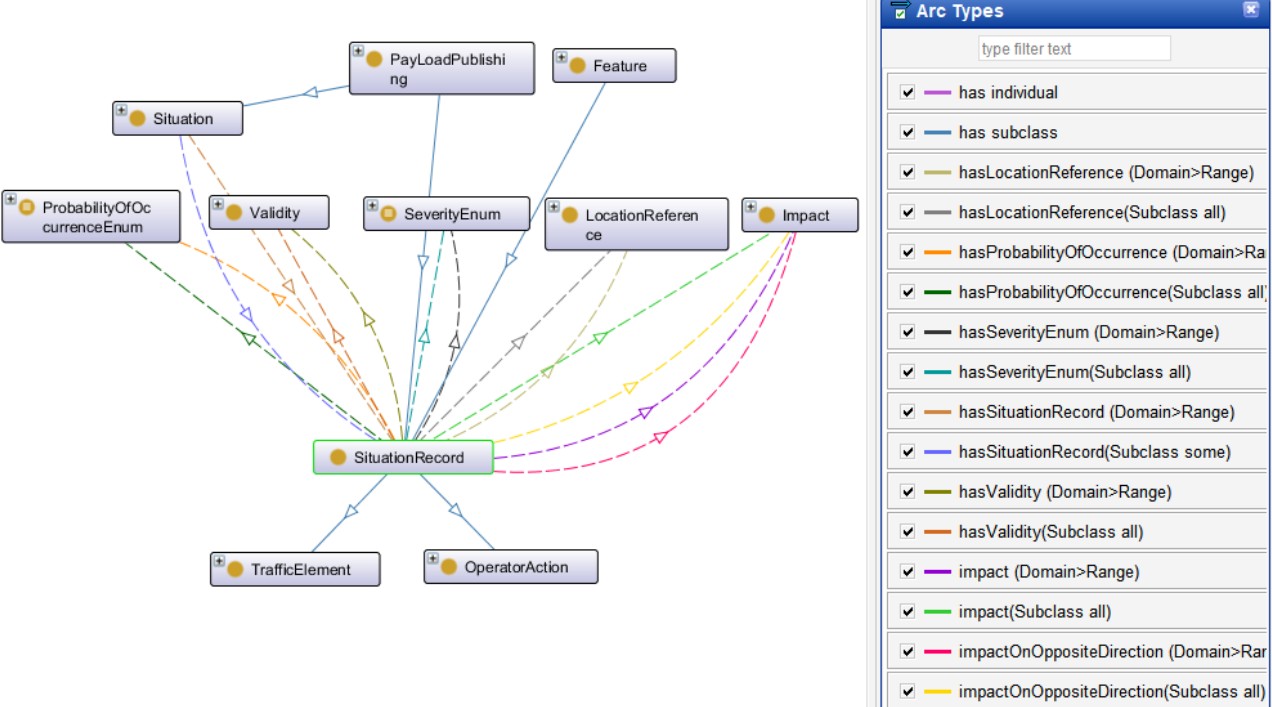

**Figure 1.** Conceptual representation of the SituationRecord and its relationships (caption in Protégé-Ontograf).

It is important highlight that the initial ontology was expanded to incorporate the following non-SRTI elements:

- Elements associated with a *SituationRecord*, including *Severity*, *Source*, and *Impact*.
- New items encompassing both old SRTI enumerations and newly introduced non-SRTI enumerations.
- The inclusion of the Mobility concept.

To address the development of a traffic ontology, the definition of interconnected subdomains has been proposed. In this context, the relevant subdomains that have been defined are as follows:

- Roads' types and features (Freeway, Toll Road, Connector, Access, Ring Road, etc.)
- Administrative Units (Towns, Countries, etc.)

Focusing on this profile SRTI, the central class of the model is the "SituationRecord" concept, along with its various attributes and relationships. The requirement stage of ontology development resulted in the identification of several scenarios and associated Competency Questions (CQs).

(a) CQs on the SituationRecord.
(b) CQs on the road catalogue.
(c) CQs on the data of administrative units.

The following are some of the CQs which were considered:

- List of events (SituationRecord) of each Situation and their attributes related to locations: road attributes, administrative units, or geographical coordinates in WGS84 format and WKT.
- According to different types of events (Conditions, Roadworks, Abnormal traffic etc.) obtain their probability of occurrence, period of validity, etc.
- Obtain all information related to administrative areas responsible to manage each event.
- Obtain situation records whose period of validity (OverallStartPeriod in DATEX II) starts before or after a given instant or whose validity period is within a range.
- Characteristics from the longest road sections from Autonomous Communities sections, whose lengths are greater than a given length, and the road and its features to which they belong.
- Obtain campsites, hostels, points of interest, national inns, train stations, hospitals, petrol stations, etc., around where the traffic incident occurs.
- Others.

Detailed documentation of the proposed Linked Open Traffic Data Model can be found at [14]. This ontology enables the definition of concepts related to road traffic, their corresponding situations, and their individual elements as outlined in the DATEX II standard.

Next, let us explore various aspects and how they were addressed through the incorporation of key vocabularies.

### 4.1. Temporal Considerations

In the DATEX II model, particularly within the SRTI profile, time plays a significant role, and several elements are associated with it. This section describes how temporal aspects have been addressed in the semantic model. Traffic events have a specific time duration, and the SRTI profile specifically focuses on non-recurrent events. These events are not scheduled in advance and do not repeat.

In DATEX II standard, *Validity* constitutes a sub-model expressly crafted to delineate the temporal validity of a situation element (e.g., as described in a *SituationRecord* instance) or the ramifications of said situation element. Within this framework, validity pertains to the timeframe in which the real-world event, activity, action, or impact being delineated transpires or is forecasted to occur. Establishing the validity period holds paramount significance, playing a pivotal role in providing real-time information services for traffic. This is particularly crucial for users of road networks who need to know, for example, whether a construction, congestion, or accident is still active, enhancing their ability to make informed decisions.

In the minimal SRTI profile, the *validity* concept, along with its associations and attributes, has been simplified, as depicted in Tables 1 and 2.

**Table 1.** Associations of the "Validity" package. Source "4b Classes and enumerations SRTI profile" document.

| Class Name | Validity |
|---|---|
| Association End | validityTimeSpecification |
| Designation | Validity time specification |
| Definition | Detailed specification for defining periods of validity, which are determined by the overall bounding start and end times. Additionally, the model addresses the scenario where valid periods may intersect with exception periods, whereby the exception periods override the validity. |
| Multiplicity | 1..1 |
| Target | OverallPeriod |

These tables demonstrate that the association with the *Period* class from the full DATEX II model has been removed. Additionally, the *overallEndTime* attribute for the *OverallPeriod* has been eliminated. However, in our model, we have chosen to retain this attribute, as it is utilized in the full DATEX model to specify the end of a period if necessary (with a multiplicity of 0..1). Furthermore, we have adopted the W3C TIME ontology [26] for temporal properties, as recommended by NTI-RISP. This ontology allows us to define properties with values related to temporal aspects.

**Table 2.** Features of the "Validity" package. Source "4b Classes and enumerations SRTI profile" document.

| Class Name | OverallPeriod |
|---|---|
| Attribute End | overallStartTime |
| Designation | Validity time specification |
| Definition | Represents the specific date and time that marks the beginning of the bounding period of validity. |
| Multiplicity | 1..1 |
| Type | DateTime |

Figure 2 illustrates a portion of the UML schema related to validity in the DATEX II 3.2 model.

Regarding the DATEX Model version 3.2, the properties within the Overall Period had specific data type restrictions, as follows:

- *overallEndTime max 1 xsd:dateTime*.
- *overallStartTime exactly 1 xsd:dateTime*.

To specify these restrictions within the model, several steps were taken. Firstly, the TIME W3C ontology [26] was imported, taking into consideration the indications provided by [38] regarding intervals: "Proper intervals are intervals whose extremes are different. Among other things, this allows using standard interval calculus and defining relations between intervals." By importing W3C TIME, the *dtx_srti:OverallPeriod* was defined as a Temporal Entity, specifically a *time:ProperInterval*.

By designating the *dtx_srti:OverallPeriod* as a *time:ProperInterval*, instances of it were created with the properties *time:hasBeginning* and *time:hasEnd*, which have a range of time:Instant. These properties accurately specify the beginning and end of the validity period for a Situation Record (dtx srti:SituationRecord). The related definitions in our model are presented in Listing 1.

Furthermore, in accordance with the W3C TIME Ontology, we can apply various interval relations between time periods within our instances if needed. Figure 3 depicts the dtx_srti:Validity class and its relationships using the Protégé software tool.

Figure 4 illustrates how the property *hasValidity* allows for the establishment of a validity interval for a *SituationRecord*. In addition, Listing 2 presents an example of a partial representation of an instance, considering a *SituationRecord* with a specified validity interval.

For this, the use of terms from the W3C TIME ontology is crucial. For example, the term "Instant" will allow us to specify the beginning and end of any interval, as specified for the *dtx_srti:Validity* concept (Listing 1).

The *SituationRecord* specification offers a robust framework and provides the capability to query the model using SPARQL, allowing us to retrieve information based on the following criteria:

- Retrieve information about a *SituationRecord* with an *OverallPeriod* that starts at the time "2022-01-04 T 08:27:04.963 + 01:00" (refer to Listing 3).
- Retrieve information about a *SituationRecords* that started after a specific date, indicated by an *OverallStartPeriod* greater than "2021-01-04 T 08:26:04.963 + 01:00". The results should be sorted in ascending order by the *OverallStartRecord* (refer to Listing 4).

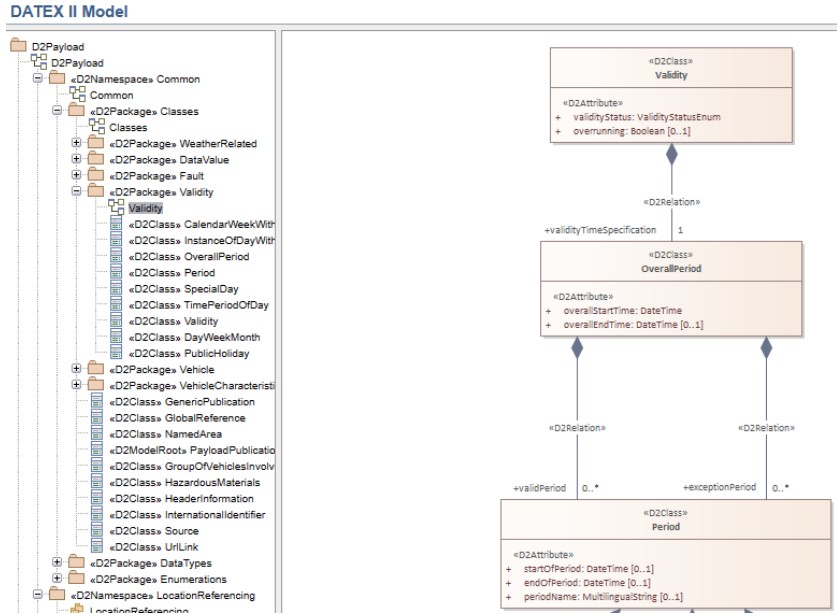

**Figure 2.** *Validity* Class. Source DATEX II Model (datex2.eu).

**Listing 1.** Definitions and relationships for dtx_srti:Validity concept.

```
##Validity
dtx_srti:Validity dtx_srti:validityTimeSpecification exactly 1 dtx_srti:OverallPeriod
dtx_srti:Validity dtx_srti:hasValidityStatus exactly 1 dtx_srti:ValidityStatusEnum

##OverallPeriod
dtx_srti:OverallPeriod rdfs:subClassOf time:ProperInterval
dtx_srti:OverallPeriod rdfs:subClassOf time:hasBeginning exactly 1 time:Instant
dtx_srti:OverallPeriod rdfs:subClassOf time:hasEnd max 1 time:Instant

## ValidityStatusEnum
dtx_srti:ValidityStatusEnum rdfs:subClassOf dtx_srti:PayLoadEnumerations
dtx_srti: ValidityStatusEnum owl:equivalentTo {active , definedByValidityTimeSpec ,
    planned , suspended}
```

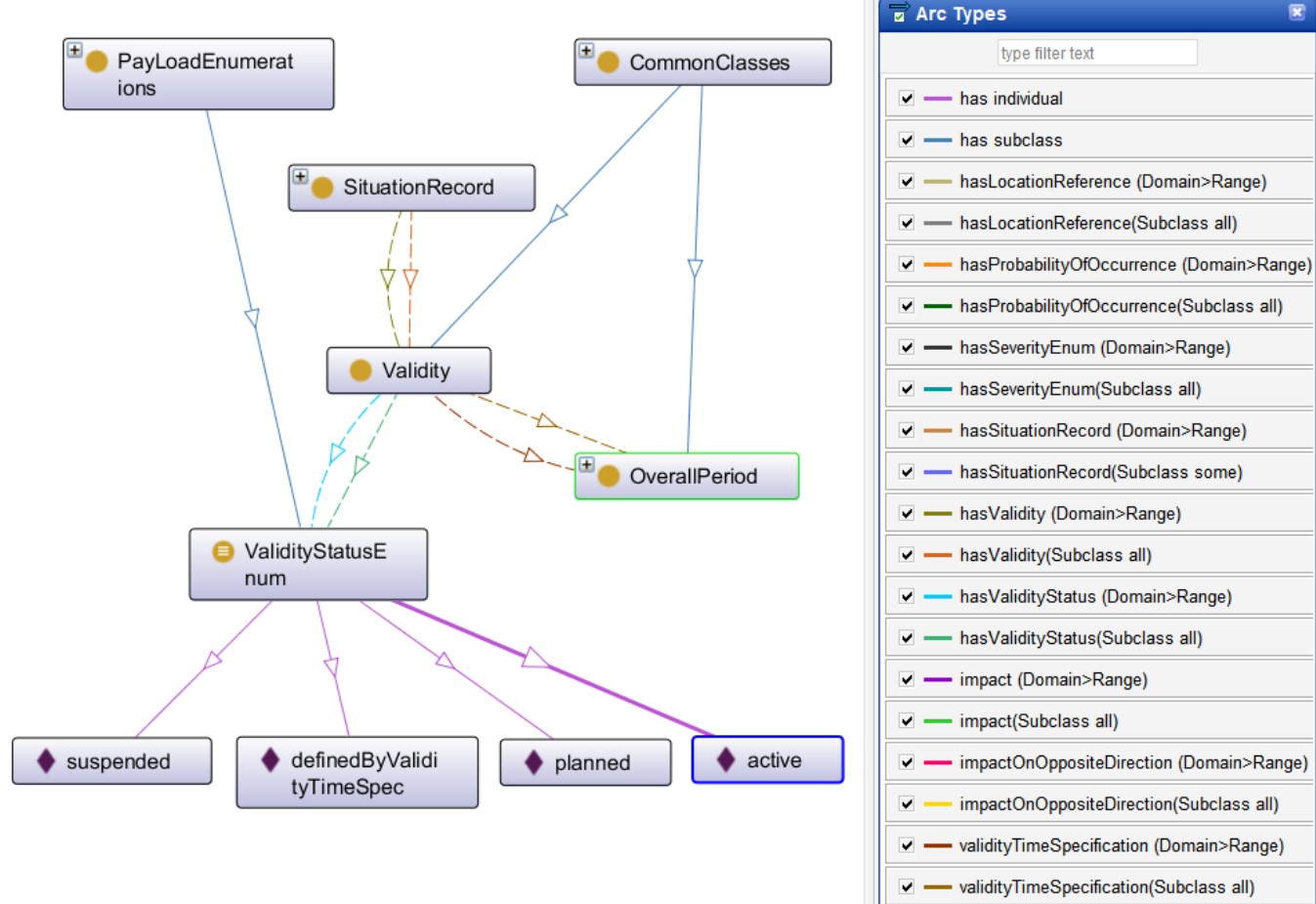

**Figure 3.** Representation of the Validity Class and its relationships in the dtx_srti main ontology (Caption in Protégé-Ontograf).

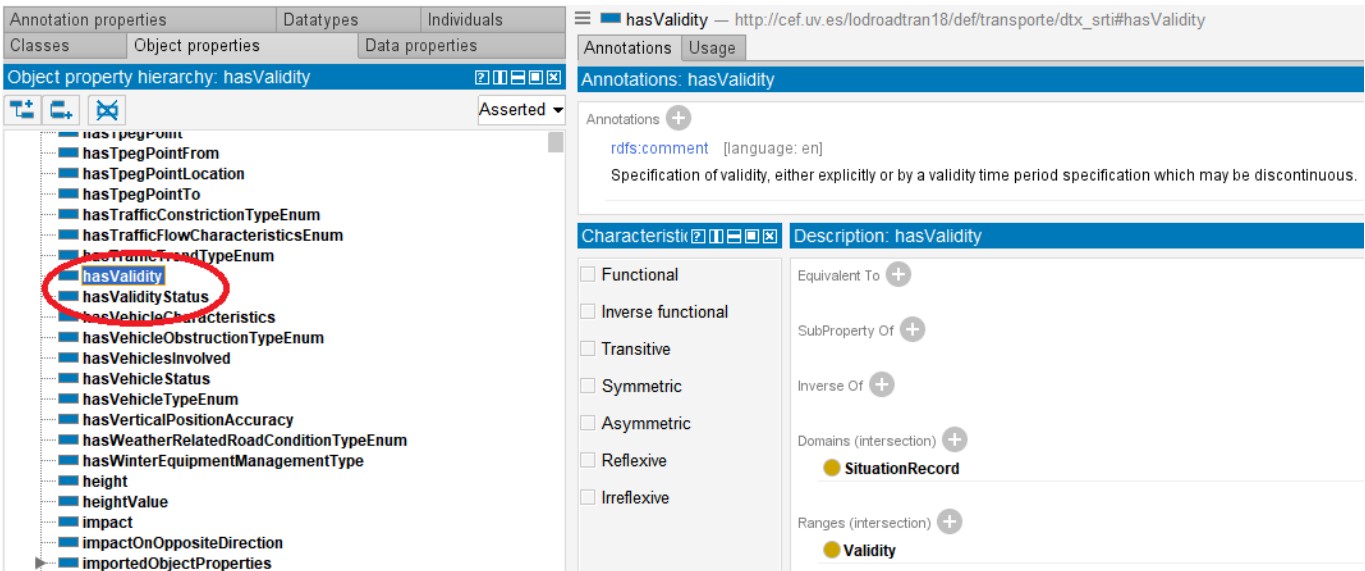

**Figure 4.** Representation of the hasValidity object property and its relationships in the dtx_srti core ontology.

**Listing 2.** SituationRecord Individual with validity intervals.

```
<dtx_srti:hasValidity>
<dtx_srti:Validity>
<dtx_srti:validityTimeSpecification>
  <dtx_srti:OverallPeriod>
     <time:hasEnd>
<time:Instant>
<time:inXSDDateTime
 rdf:datatype="http://www.w3.org/2001/XMLSchema#dateTime">
   2021-11-30T17:00:00.019Z
</time:inXSDDateTime>
</time:Instant>
     </time:hasEnd>
     <time:hasBeginning>
<time:Instant>
<time:inXSDDateTime
 rdf:datatype="http://www.w3.org/2001/XMLSchema#dateTime">
   2021-11-30T07:00:00.019Z
</time:inXSDDateTime>
</time:Instant>
     </time:hasBeginning>
  </dtx_srti:OverallPeriod>
</dtx_srti:validityTimeSpecification>
</dtx_srti:Validity>
</dtx_srti:hasValidity>
```

**Listing 3.** TIME SPARQL query 1.

```
PREFIX rdf: <http://www.w3.org/1999/02/22-rdf-syntax-ns#>
PREFIX owl: <http://www.w3.org/2002/07/owl#>
PREFIX rdfs: <http://www.w3.org/2000/01/rdf-schema#>
PREFIX xsd: <http://www.w3.org/2001/XMLSchema#>
PREFIX dtx_srti: <http://cef.uv.es/lodtran18/def/transporte/dtx_srti#>
PREFIX time: http://www.w3.org/2006/time/
SELECT DISTINCT ?situationRecord ?PO ?Validity ?SRM ?CT ?FSVT ?V ?VT ?OverallPeriod ?
     instant
WHERE {
?situationRecord a dtx_srti:SituationRecord;
    dtx_srti:hasProbabilityOfOccurrence ?PO;
    dtx_srti:hasValidity ?Validity;
    dtx_srti:safetyRelatedMessage ?SRM;
    dtx_srti:situationRecordCreationTime ?CT;
    dtx_srti:situationRecordFirstSupplierVersionTime ?FSVT;
    dtx_srti:situationRecordVersion ?V;
    dtx_srti:situationRecordVersionTime ?VT.
 ?Validity dtx_srti:validityTimeSpecification ?OverallPeriod .

 ?OverallPeriod time:hasBeginning ?instant .
 ?instant time:inXSDDateTime "2022-01-04T08:27:04.963+01:00"^^xsd:dateTime }
```

**Listing 4.** TIME SPARQL query 2.

```
PREFIX rdf: <http://www.w3.org/1999/02/22-rdf-syntax-ns#>
PREFIX owl: <http://www.w3.org/2002/07/owl#>
PREFIX rdfs: <http://www.w3.org/2000/01/rdf-schema#>
PREFIX xsd: <http://www.w3.org/2001/XMLSchema#>
PREFIX dtx_srti: <http://cef.uv.es/lodtran18/def/transporte/dtx_srti#>
PREFIX time: <http://www.w3.org/2006/time/>
 SELECT distinct ?situationRecord ?ProbabilityOfOccurrence ?safetyRelatedMessage ?
     CreationTime ?FirstSupplierVersionTime ?Version ?VersionTime ?OverallStartPeriod
WHERE {
?situationRecord a dtx_srti:SituationRecord;
   dtx_srti:hasProbabilityOfOccurrence ?ProbabilityOfOccurrence;
   dtx_srti:hasValidity ?Validity;
   dtx_srti:safetyRelatedMessage ?safetyRelatedMessage;
   dtx_srti:situationRecordCreationTime ?CreationTime;
   dtx_srti:situationRecordFirstSupplierVersionTime ?FirstSupplierVersionTime;
   dtx_srti:situationRecordVersion ?Version;
   dtx_srti:situationRecordVersionTime ?VersionTime.
?Validity dtx_srti:validityTimeSpecification ?OverallPeriod .
?OverallPeriod time:hasBeginning ?instant .
?instant time:inXSDDateTime ?OverallStartPeriod

FILTER (?OverallStartPeriod > "2021-01-04T08:26:04.963+01:00"^^xsd:dateTime). }
ORDER BY ?OverallStartPeriod
```

*4.2. Reusing QUDT to Implement Units of Measure*

In the DATEX II standard, a highly significant package known as the DataValue package exists, which is designed to describe data values of measurable or calculable entities. This package encompasses types for data values and supplements them with additional quality and error information. Additionally, beyond the basic datatypes, specific datatypes are available, such as AngleInDegrees, MetresAsFloat, Percentage, etc. Given the proliferation of stable vocabularies related to units of measure and their importance in the model, this section aims to address certain aspects of their specification.

In order to express units of measure, we conducted a review of the main vocabularies pertaining to this topic, including the Ontology of units of Measure (OM) [39], Quantities, Units, Dimensions and Data Types (QUDT) [40], Custom Data Type (CDT) [41], and others. After careful consideration, we selected the QUDT vocabulary. QUDT consists of a collection of vocabularies that represent various quantity and unit standards, and it offers solutions to problems in this domain. We utilized the QUDT ontology, specifically focusing on the class *qudt:QuantityValue*, which is defined as a value representing a quantity numerically in relation to a chosen unit of measure.

Additionally, the *qudt:unit* is an *ObjectProperty* with a range of *qudt:Unit*. It serves as a reference to the unit of measure associated with a quantity of interest, whether it is a variable or a constant. The concept of *qudt:QuantityValue* imposes a restriction on the "unit" property, allowing it to have exactly one unique value from the general concept *owl:Thing*. Moreover, the vocabulary *qudt-unit* [42] contains numerous units of measure, each of which is a subtype of *qudt:Unit*. For instance, Listing 5 illustrates the specification of the *unit:KiloM* (kilometre) unit.

Thus, in our model, the new concepts related to units of measure are subclasses of *qudt:QuantityValue*. They also impose similar restrictions on the *qudt:unit* property, but with specific ranges that correspond to individual values (units of measure from *qudt:unit*) instead of the general class *owl:Thing*. Additionally, these concepts have a *numericValue* data type property with a multiplicity restriction and a range of XSD (XML Schema Definition) types, in accordance with the DATEX II model, as depicted in Figure 5 and Table 3.

The subsequent step involves importing the individuals from the *qudt* vocabulary using the prefix *qudt:unit*. Specifically, we import the units "Meter", "Degree", "Kilometer per Hour", and "Kilometer."

Taking all the aforementioned factors into consideration, we have developed various concepts pertaining to units of measure in our model.

**Listing 5.** qudt–unit:KiloM specification.

```
unit:KiloM
  a qudt:DerivedUnit ;
  a qudt:Unit ;
  dcterms:description "A␣common␣metric␣unit␣of␣length␣or␣distance.␣One␣kilometer␣equals␣
      exactly␣1000␣meters,␣about␣0.621␣371␣19␣mile,␣1093.6133␣yards,␣or␣3280.8399␣feet.␣
      Oddly,␣higher␣multiples␣of␣the␣meter␣are␣rarely␣used;␣even␣the␣distances␣to␣the␣
      farthest␣galaxies␣are␣usually␣measured␣in␣kilometers.␣"^^rdf:HTML ;
  qudt:allowedUnitOfSystem sou:CGS-EMU ;
  qudt:allowedUnitOfSystem sou:CGS-GAUSS ;
  qudt:allowedUnitOfSystem sou:SI ;
  qudt:conversionMultiplier 1000.0 ;
  qudt:dbpediaMatch "http://dbpedia.org/resource/Kilometre"^^xsd:anyURI ;
  qudt:hasDimensionVector qkdv:A0E0L1I0M0H0T0D0 ;
  qudt:hasQuantityKind quantitykind:Length ;
  qudt:informativeReference "http://en.wikipedia.org/wiki/Kilometre?oldid=494821851"^^xsd:
      anyURI ;
  qudt:isScalingOf unit:M ;
  qudt:prefix prefix:Kilo ;
  qudt:symbol "km" ;
  qudt:ucumCode "km"^^qudt:UCUMcs ;
  qudt:uneceCommonCode "KMT" ;
  qudt:unitOfSystem sou:SI ;
  rdfs:isDefinedBy <http://qudt.org/2.1/vocab/unit> ;
  rdfs:label "Kilometer"@en-us ;
  rdfs:label "Kilometre"@en ;
```

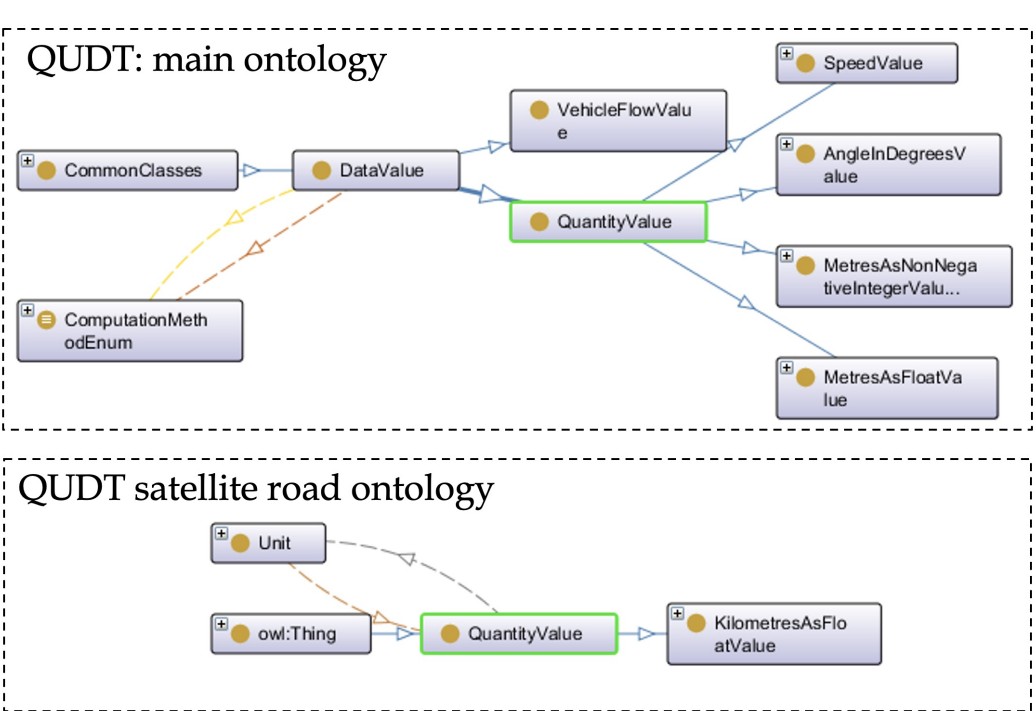

**Figure 5.** New classes based on QUDT in the main ontology dtx_srti. (top box) and in the satellite roads ontology (bottom box). Colors identify the different relation types between concepts.

**Table 3.** List of imported individuals from QUDT (qudt-unit).

| UoM | URI with Prefix | Used In: | |
|---|---|---|---|
| http://qudt.org/vocab/unit (accessed on 12 June 2023) | | Ontology | LClasses |
| Meter | qudt-unit:M | dtx-srti | MetresAsFloatValue , MetresAsNonNegativeIntegerValue |
| Degree | qudt-unit:DEG | dtx-srti | AngleInDegrees |
| Kilometer per Hour | qudt-unit:KiloM-PER-HR | dtx-srti | SpeedValue |
| Kilometer | qudt-unit:KiloM | roads | KilometresAsFloatValue |

Figure 6 depicts the description of the newly introduced classes, based on the QUDT vocabulary. Each class is associated with two different properties. The first property, *numericValue*, has a range that corresponds to a specific XSD type, while the second property, *unit*, refers to an individual from the *qudt* vocabulary.

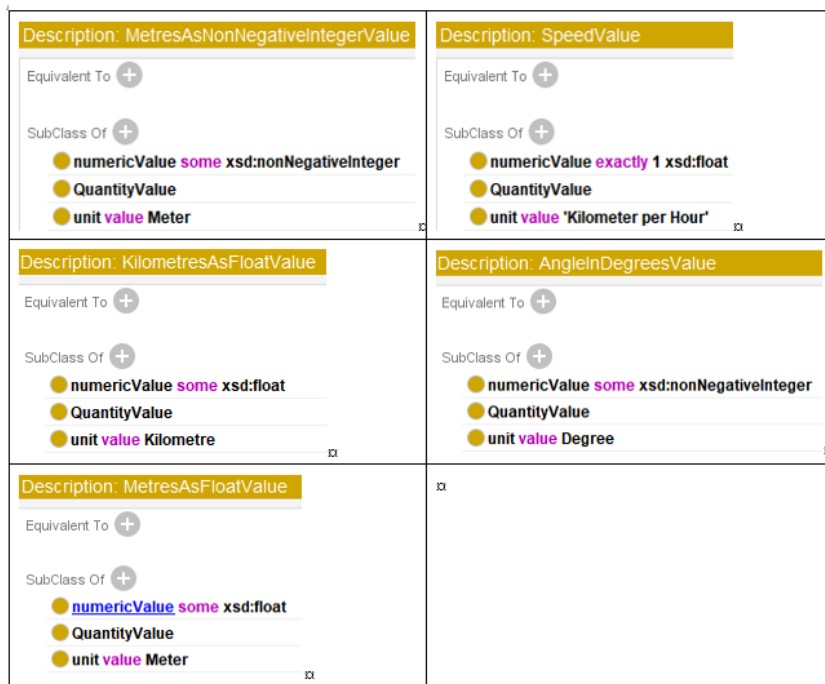

**Figure 6.** Description of the new classes in the semantic model, which are based on the QUDT vocabulary.

In Listing 6, a partial instance of a *SituationRecord* is presented, demonstrating the usage of the *qudt:numericValue* and *qudt:unit*" properties to define a specific distance of 90.0 m.

**Listing 6.** SituationRecord Individual with Quantity Value.

```
<dtx_srti:DistanceFromLinearElementStart>
            <dtx_srti:distanceAlong>
              <qudt:QuantityValue>
                <qudt:numericValue rdf:datatype=http://www.w3.org/2001/XMLSchema#
                    float> 90.0
              </qudt:numericValue>
                <qudt:unit rdf:resource="http://qudt.org/vocab/unit#M"/>
              </qudt:QuantityValue>
            </dtx_srti:distanceAlong>
    </dtx_srti:DistanceFromLinearElementStart>
```

### 4.3. Geographical Aspect for Reusing GeoSPARQL

In the DATEX II model, locations are typically expressed using point coordinates (latitude and longitude). Therefore, it is of paramount importance to include GeoSPARQL to efficiently extract geographical information and establish relationships between objects

In [43], the OGC GeoSPARQL is introduced as a vocabulary that enables the representation of geospatial data in RDF. It extends the SPARQL query language to handle geospatial data and is designed to accommodate both qualitative spatial reasoning and quantitative spatial computations.

To express geographical features using this vocabulary, we imported the GeoSPARQL ontology. In our model, the *dtx_srti:SituationRecord* class was defined as a subclass of *subClassOf geosparql:Feature*. This inheritance relationship allows us to utilize the *hasGeometry* property for our concepts. Figures 7 and 8 depict the usage of the *geosparql: Feature* in our model and the subclass relationship with the *SituationRecord* concept.

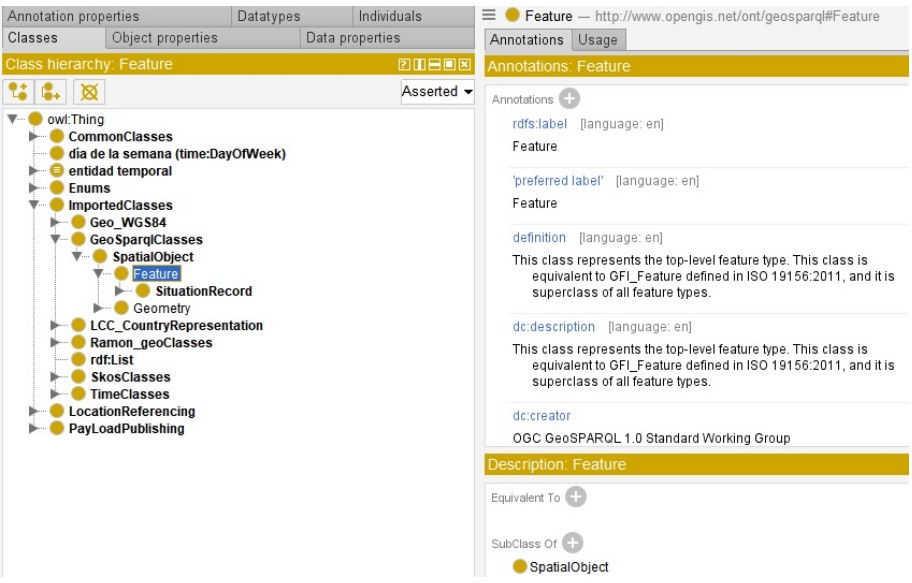

**Figure 7.** Use of geosparql:Feature.

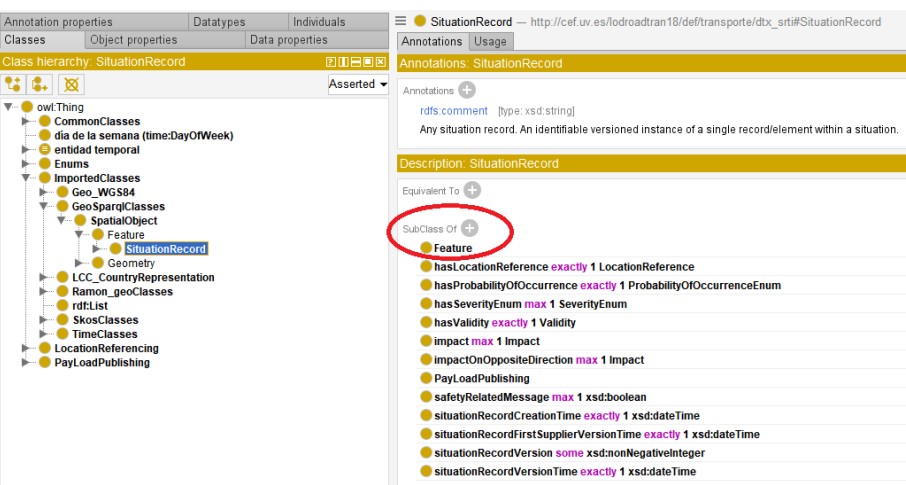

**Figure 8.** Definition of dtx_srti:SituationRecord using a subClassOf relationship with geosparql:Feature.

Listing 7 provides an example of an instance of a *SituationRecord*, demonstrating how a geometry feature can be added to it.

**Listing 7.** SituationRecord individual with geometry property.

```
<owl:NamedIndividual rdf:about="http://cef.uv.es/lodtran18/def/transporte/dtx_srti#
    SituationRecord/GUID_Suc_612387_612387">
        <rdf:type rdf:resource="http://cef.uv.es/lodtran18/def/transporte/dtx_srti#
            SituationRecord"/>

        <geosparql:hasGeometry rdf:resource="http://cef.uv.es/lodtran18/def/transporte/dtx_
            srti#geoGUID_Suc_612387_612387"/>
</owl:NamedIndividual>

<owl:NamedIndividual rdf:about="#geoGUID_Suc_612387_612387">
        <rdf:type rdf:resource="http://www.opengis.net/ont/sf#LineString"/>
        <geosparql:asWKT rdf:datatype="http://www.opengis.net/ont/geosparql#wktLiteral">
            LINESTRING(-3.02388881 37.101178705838,-3.02316586 37.100498995838)</geosparql:
            asWKT>
</owl:NamedIndividual>
```

### 4.4. Reusing Other Vocabularies

Other vocabularies were also used to specify several DATEX II concepts in the model. Table 4 shows these vocabularies.

**Table 4.** Ontologies used for several DATEX II concepts.

| Concept Defined in DATEX II | Ontology Used for |
|---|---|
| AdministrativeAreaName | http://vocab.linkeddata.es/datosabiertos/def/sector-publico/territorio (accessed on 11 December 2023) |
| NutsNamedArea | https://ec.europa.eu/eurostat/ramon/ontologies/geographic (accessed on 11 December 2023) rdf |
| | http://data.europa.eu/nuts/code (accessed on 11 December 2023) |
| | htttp://data.europa.eu/euodp/repository/ec/estat/nuts/nuts (accessed on 11 December 2023) rdf |
| CountryCode and SubdivisionCode | http://www.omg.org/spec/LCC/Countries/Country (accessed on 11 December 2023) Representation/ |
| Language Representation | https://www.omg.org/spec/LCC/Languages/Language (accessed on 11 December 2023) Representation/ |
| PointCoordinates | http://www.w3.org/2003/01/geo/WGS84, (accessed on 11 December 2023) |
| | https://datos.ign.es/def/geo_core (accessed on 11 December 2023) |

For the *CountryCode* or *SubdivisionCode*, LCC Country Representation from OMG [44] has been used in the model. Figure 9 shows the concepts Alpha2Code, Country and CountrySubdivision imported from LCC for our model and Figure 10 the definition of the first one.

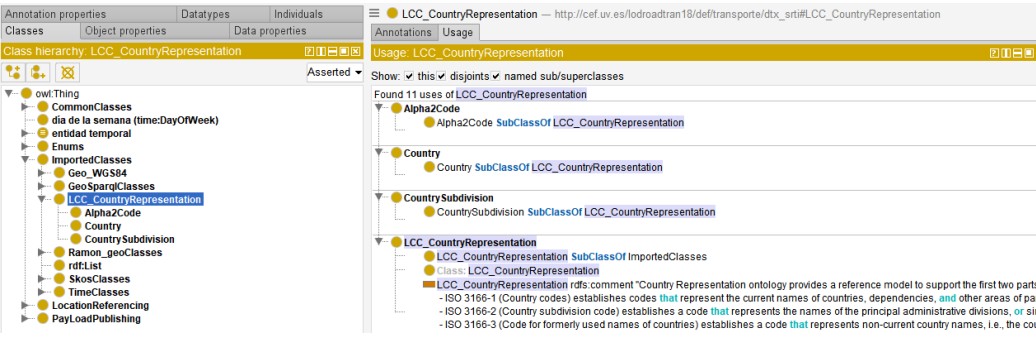

**Figure 9.** Imported "Alpha2Code" , "Country", and "CountrySubdivision" classes from LLC Country Representation.

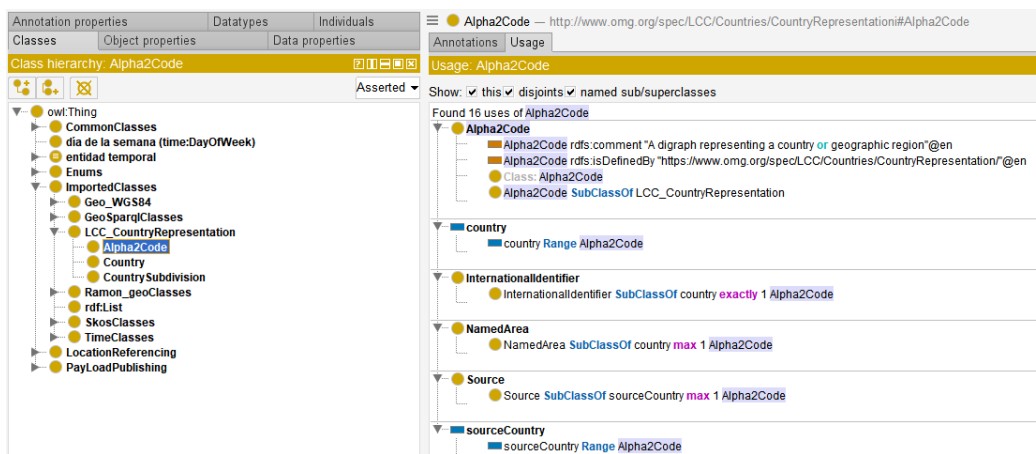

**Figure 10.** "Alpha2Code" definition.

## 5. Conversion and Validation Processes

The technical system for obtaining traffic data from different sources, tagging, and processing them as LOD, together with subscription and querying capabilities, was designed and tested in order to evaluate the ontology. The developed ontology was used as support to create instances based on it, and an interface to query the linked open data (SPARQL endpoint) was developed and manually tested.

### 5.1. Conversion Process

All the development of the conversion process from XML to RDF formats has been based on Eclipse using the JAXB and JENA libraries. The Java™ Architecture for XML Binding (JAXB) provides an API and tools that automate the mapping between XML documents and Java objects. During development, JAXB has been used to convert the NAP XML files to JAVA objects for easy data handling (see Figure 11).

```java
@XmlRootElement(name = "linearElement", namespace="http://datex2.eu/schema/3/locationReferencing")
@XmlAccessorType(XmlAccessType.NONE)
@XmlType(name="linearElement", namespace="http://datex2.eu/schema/3/locationReferencing")

public class LinearElement {

    String roadName;
    String roadNumber;

    public String getRoadName() {
        return roadName;
    }
    @XmlElement(name = "roadName", namespace="http://datex2.eu/schema/3/locationReferencing")
    public void setRoadName(String roadName) {
        this.roadName = roadName;
    }
    public String getRoadNumber() {
        return roadNumber;
    }

    @XmlElement(name = "roadNumber", namespace="http://datex2.eu/schema/3/locationReferencing")
    public void setRoadNumber(String roadNumber) {
        this.roadNumber = roadNumber;
    }

}
```

**Figure 11.** Linear Element JAXB Class.

In Figure 12, a snippet segment of the LOAD_DATEX_DATA classis presented. It allows conversion from XML to JAVA classes using Unmarshall.

```
public class LoadDatexData {

    public SituationPublication situationPublication = null;
    final static Logger logger = Logger.getLogger(LoadDatexData.class);

    public SituationPublication LoadData() throws JAXBException

    // Incluimos las diferentes clases para el Unmarshall

    JAXBContext jc = JAXBContext.newInstance(PayloadPublication.class, SituationPublication.class, Situation.class, SituationRecord.class,
            OperatorAction.class, NetworkManagement.class,TrafficElement.class, Conditions.class, RoadSurfaceConditions.class, WeatherRelatedRoadConditions.class,
            OverallPeriod.class, LocationReference.class, PointLocation.class, DistanceFromLinearElementStart.class, DistanceAlongLinearElement.class
            , TpegPointLocation.class, TpegSimplePoint.class, TpegPoint.class, TpegJunction.class,TpegNonJunctionPoint.class, TpegDescriptor.class, TpegOtherPointDescriptor.class,
            TpegPointDescriptor.class, DescriptorValues.class, LinearLocation.class, SingleRoadLinearLocation.class, LinearWithinLinearElement.class, AbnormalTraffic.class
            ,TpegLinearLocation.class, ConstructionWorks.class, MaintenanceWorks.class, AdministrativeAreaOfLinearSection.class, AnimalPresenceObstruction.class, Obstruction.class, EnvironmentalObstruction.
            ,GeneralObstruction.class, VehicleObstruction.class, NonWeatherRelatedRoadConditions.class, PoorEnvironmentConditions.class, Delays.class, Impact.class
            ,AlertCPoint.class, AlertCMethod4Point.class, AlertCDirection.class, AlertCMethod4PrimaryPointLocation.class, OffsetDistance.class
            ,AlertCLinear.class, AlertCMethod4Linear.class, AlertCMethod4SecondaryPointLocation.class,
            Source.class, Mobility.class, Roadworks.class, MaintenanceWorks.class, Subjects.class, HeaderInformation.class, InternationalIdentifier.class, FeedDescription.class
            , LocationGroupByList.class, LocationGroup.class, LocationGroupByReference.class, AreaDestination.class, AreaLocation.class, Payload.class);

    Unmarshaller unmarshaller = jc.createUnmarshaller();
```

**Figure 12.** Unmarshall DATEX II xml file.

During development, Java classes have been created using JENA to read the ontology and create content in RDF formats (see Figure 13).

```
public class PointAlongLinearElementSem {

    ObjectProperty hasAdministrativeAreaOfPoint;
    ObjectProperty directionRelativeAtPoint; //directionRelativeAtPoint
    ObjectProperty hasLinearElement;
    ObjectProperty hasDistanceAlongLinearElement;

    // Semantic Classes
    DistanceFromLinearElementStartSem distanceFromLinearElementStartSem;
    LinearElementSem linearElementSem;

    // Ontology Classes
    OntClass pointAlongLinearElementClassOnto;

    // Individual
    Individual directionEnumIndv;

    // Resource
    Resource resourcePointAlongLinearElement;

    // Statement
    Statement stmtLinearDirectionEnum;
    Statement stmtDistanceFromLinearElementStart;
    Statement stmthasLinearElement;
    Statement stmtHasAdministrativeAreaOfPoint;

    public PointAlongLinearElementSem(OntModel base, String namespace, OntModel baseCarretera, OntModel baseUnidadesAdm,String namespaceCarretera, String namespaceUnidadesAdm, SituationRecord situationRecord) {

        this.hasAdministrativeAreaOfPoint = base.getObjectProperty(namespace+"hasAdministrativeAreaOfPoint");
        this.directionRelativeAtPoint = base.getObjectProperty(namespace+"directionRelativeAtPoint");
        this.hasLinearElement = base.getObjectProperty(namespace+"hasLinearElement");
        this.hasDistanceAlongLinearElement = base.getObjectProperty(namespace+"hasDistanceAlongLinearElement");
        this.pointAlongLinearElementClassOnto = base.getOntClass( namespace + "PointAlongLinearElement" );

        // Point Location
        PointLocation pointLocation = (PointLocation)situationRecord.getLocationReference();
        pointLocation.getSupplementaryPositionalDescription();

        // Clases Semanticas
        distanceFromLinearElementStartSem = new DistanceFromLinearElementStartSem( base, namespace, situationRecord);
        linearElementSem = new LinearElementSem( base, baseCarretera, namespace, namespaceCarretera, situationRecord);
```

**Figure 13.** PointAlongLinearElement JENA Class.

## 5.2. RDF Validator and OOPS! (OntOlogy Pitfall Scanner!)

To detect anomalies in the ontologies, at different stages of development, the online anomaly detection tools were used. Firstly, in order to verify the accuracy of the RDF specification, we employed the W3C validation tool (See Figure 14).

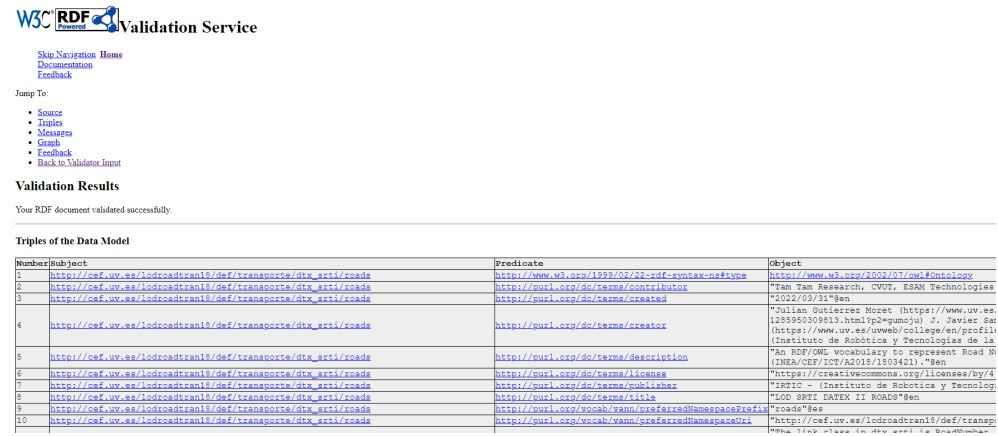

**Figure 14.** W3C validation service.

Upon completion of the model, we proceeded to utilize the OOPS! tool. It is noteworthy that this tool identifies errors of minimal significance and, in our opinion, some false positives classified as major errors, which we will elaborate on below (see Figure 15).

In the model, all terms, attributes, and relationships deemed significant were appropriately annotated with their descriptions and other relevant information. That is, the

information available in DATEX II XML was transferred to the model. However, a substantial number of properties, etc., within the model do not necessitate such information, and therefore lack annotations.

Simultaneously, we made the decision to incorporate certain properties whose unique purpose was to organize the model, particularly to segregate imported concepts. These organizational properties neither have a domain nor a range. Furthermore, it is worth mentioning that the tool identifies some errors of the same kind related to imported ontologies, which are not inherent to our model.

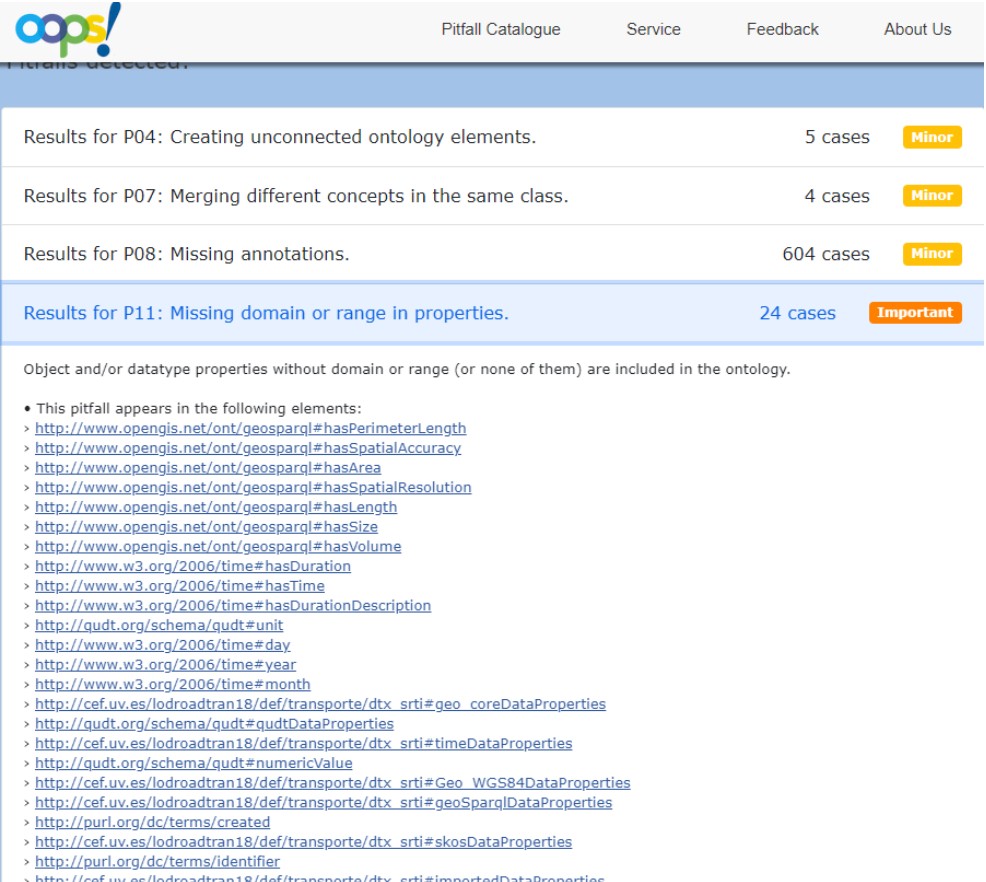

**Figure 15.** Ontology Pitfall Scanner web page caption.

*5.3. SPARQL Interface*

A SPARQL Interface was designed to allow direct and federated queries against data stored in a Virtuoso Open Server Endpoint. Several example queries were provided to ease the use of the Interface by users who are not yet proficient with the SPARQL language.

The design and preparation of the queries was carried out according to the process of instantiation and addition of new elements to the model. Currently, the active interface is available at the Spanish national access point [45]. Several tests were carried out through the interface using different queries (according to the Competency Questions) on the data of traffic related to incidents, administrative units and roads.

1.  Verification of the local consultation and federation capabilities: The development and deployment of functionalities within the Spanish NAP, which allow the publication of standardized metadata files in several formats and specifications, mandatory to achieve the federation with other open data portals like the Spanish Open Data Portal [46] and the European Open Data Portal [47]. Figures 16 and 17 present two of the queries used for testing purposes. In addition, Figure 17 shows how federated

queries link the data generated by LOD Converter with other SPARQL Endpoints, such as those presented in IGN [48].

2. LOD View tests: Various classes from DATEX II, such as Situations and Situation Records, were tested. Instances related to roads and administrative units were also tested and linked. For this purpose, LodView [49] was utilized. LodView, in conjunction with a SPARQL endpoint, allows the publication of RDF data in compliance with LOD standards. It is a Java web application based on Spring and Jena, providing W3C standard-compliant IRI dereferencing. LodView is easily configurable and deployable for developers, significantly enhancing the end user's experience when accessing HTML representations of RDF resources. Figure 18 depicts a SituationRecord resource using LodView.

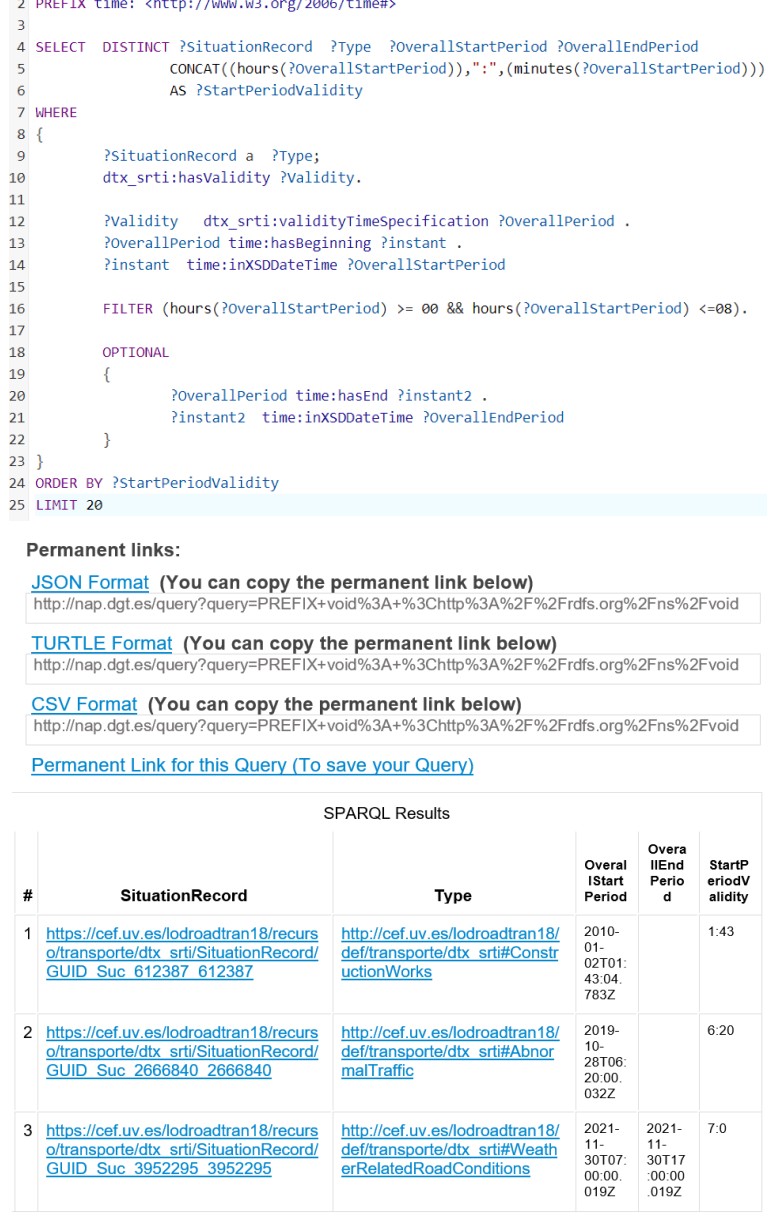

**Figure 16.** SPARQL Interface. Query and results to find all those events and their type, whose validity period began between 12 midnight and 9 a.m.

```
1  PREFIX btn100:<https://datos.ign.es/def/btn100#>
2  PREFIX dtx_srti: <http://cef.uv.es/lodroadtran18/def/transporte/dtx_srti#>
3  PREFIX geosparql: <http://www.opengis.net/ont/geosparql#>
4  PREFIX dc: <http://purl.org/dc/terms/>
5
6  SELECT DISTINCT ?situationRecord ?uriHospital ?name bif:st_distance( ?geoHospitalLocali,?asWKT ) AS ?Distan
7
8  WHERE
9      {
10          ?situationRecord a dtx_srti:WeatherRelatedRoadConditions;
11          geosparql:hasGeometry ?Geometry.
12          ?Geometry geosparql:asWKT ?asWKT.
13
14          SERVICE <https://datos.ign.es/sparql>
15          {
16              ?uriHospital a btn100:Hospital ;
17              dc:title ?name ;
18              geosparql:hasGeometry ?geoHospital .
19              ?geoHospital geosparql:asWKT ?geoHospitalLocali .
20
21          }
22          FILTER (bif:st_intersects(?geoHospitalLocali, ?asWKT,5))
23      }
```

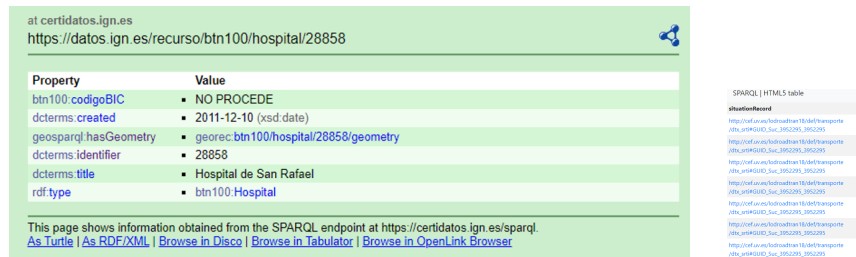
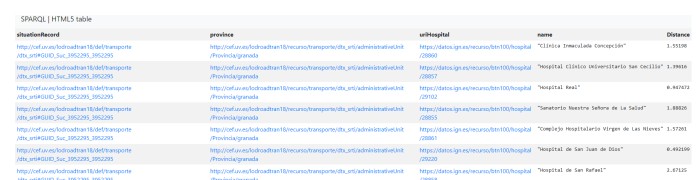

**Figure 17.** SPARQL Interface. Query: Hospitals less than 5 km away, specifying their value, from those traffic incidents of the type "WheatherRelatedRoadConditions".

**Figure 18.** Visualization of information related to a Situation Record through LodView.

In Figure 18, we can observe that by initiating from information about the administrative area to which a specific road belongs (in this case, the province of Zaragoza in Spain),

we can seamlessly access comprehensive details about the incident location. Through interaction with the ontology, automatic access to multiple linked external sources becomes possible.

Finally, the finalized models, consisting of a main ontology and two secondary ontologies, were shared and made available through our server. The ontologies and their respective URLs are as follows:

1. Ontology of DATEX II-based traffic events SRTI:
   https://cef.uv.es/lodroadtran18/def/transporte/dtx_srti/
2. Ontology of administrative units of the Spanish and Czech territory: https://cef.uv.es/lodroadtran18/def/transporte/dtx_srti/\administrativeUnit/
3. Ontology of roads of the Spanish and Czech national road network:
   https://cef.uv.es/lodroadtran18/def/transporte/dtx_srti/roads/

Moreover, the main ontology is also available via the LOV repository: https://lov.linkeddata.es/dataset/lov/vocabs/dtx_srti

## 6. Conclusions

This paper introduces the *dtx_srti* ontology for semantic modeling of road traffic data and metadata within the context of SRTI. As of the time of writing, *dtx_srti* stands as the only semantic model based on DATEX II V3.2 and SRTI profile in accordance with the ITS Directive. This model enhances the basic service of accessing open traffic data provided by the traffic National Access Point (nap.dgt.es), the Spanish Open Data Portal (datos.gob.es), and the European Data Portal (data.europa.eu). It not only facilitates data viewing and downloading but also improves the extraction of data meaning and enables other services that are only achievable using LOD.

Its main emphasis lies in the concept of SituationRecord and its attributes and relationships with other concepts. Consequently, this concept assumes the role of the primary class in the ontology. Aspects related to the event's location and the specification of its geographical coordinates in different formats become essential tools for linking with other location-based services.

To assess the model, queries were designed to verify that the various competency questions initially specified were resolved satisfactorily

The significance of having an LOD dataset for traffic information lies not only in the inclusion of data but also in the incorporation of links to other thematic datasets available on the web. This allows for federated queries that span across datasets from different platforms and sectors, not limited to just traffic data. Incorporating external data from diverse sources can add value to the tools and provide potential benefits for various applications.

The entire research, in addition to the semantic modelling partially shown in this article, has allowed the deployment of traffic information data with metadata collected and it has been made available to the public through Open Data systems and made harvestable by respective national data catalogues and the European Data Portal. Both technical and legal issues in national and European context were addressed, allowing us to tailor our development in accordance with your recommendations and specifications. The Action considered the general and technical requirements/constraints established by the European Data Portal. Once the requirements for being harvested by the European Data Portal were identified, the Action started the federation process. It serves as an integration architecture that allows interoperability between platforms.

Storing historical data (situation records) in SPARQL endpoint will help researchers to perform time and location-based searches, which will significantly improve usability of the LOD-SRTI resource. This, however, will considerably increase the amount of data and, consequently, the storage size needed and will need to be managed in future.

On the other hand, the possibilities for implementing impact measurement can be summarized in two approaches: measuring the impact of each dataset by making them visible (likes, downloads, etc.) or measuring usage through connections to a given system. The availability of tools that make possible to know the behavior of users and the level

of use of the data is necessary: to demonstrate the success of the Action; to compare the results with other systems; and potentially establish policies in the area of transport. Understanding user demands beyond compliance with the 2019 PSI Directive enables policies to be established to create high-value data and thematic data ecosystems. Also, "data sharing" between the public and private sector is recommended, respecting intellectual property and privacy.

**Author Contributions:** J.J.S.-Z.: Conceptualization, Investigation, Methodology, Formal analysis, Writing—original draft. J.G.-M.: Conceptualization, Investigation, Methodology, Software, Data curation, Writing and Review. J.M.R.: Conceptualization, Methodology, Formal analysis, Writing and Review. J.J.M.-D.: Supervision, Conceptualization, Writing and Review. V.R.T. Implementation and validation process. All authors have read and agreed to the published version of the manuscript.

**Funding:** This research received no external funding.

**Acknowledgments:** We acknowledge the HADEA Agency of the European Commission for co-financing the CEF Action Number 2018-EU-IA-0088, Grant Agreement number INEA/CEF/IC-T/A2018/1803421, titled 'Supporting the cross-border use of Road Traffic Data with Linked Open Data based on DATEX II' (LOD-RoadTran18). The results of this action laid the groundwork for the research presented in this paper.

**Conflicts of Interest:** The authors declare no conflict of interest.

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
