# Peer review of "Semantic Modelling Approach for Safety-Related Traffic Information Using DATEX II"

_information, doi:10.3390/info15010003_

Round 1
Reviewer 1 Report
Comments and Suggestions for Authors
Thank you for having your work under review, it was a pleasure to read and review. However, there are a few points that need to considered for next round of review, especially those regarding the methodological approach of engineering the ontologies (which is completely missing).
Moreover, there are a few other issues to be considered, mostly regarding the presentation and structure of your paper. Please also take care some language/syntax issues, as well as visibility of figures.
Please find the detailed comments within the attached pdf file (to ease the process of revising).

Author Response
Reviewer #1
Pag 2. Line 61: “This paper provides comprehensive details on the development of the semantic model, which consists of a main ontology and two secondary ontologies.”
modules? is there a modular OEM followed? in general? is there any OEM followed during the eng. of the ontology? if no, why?
It was explained why modularity is a crucial aspect and the need for working independently in secondary ontologies with concepts from Spain and the Czech Republic.
Pag 2. Line 67 “exposition”. description?
Thanks for your suggestion, we agree with the change to "description."
Pag 5. Line 188. a) Section 4 is titles 'development and implementation of..." which is vague having two synonyms in the same title. Perhaps you would like to say 'design and impementation' whch is reasonable. However, starting from this point, and ommiting the Specification phase of an ontology eng. lifecycle, is quite rare/strange, and not justified. Why you have followed this approach? Why the specification of the aim and requirements of the ontology are missing? Why not following the distinct phases/processes/tasks of an OEM (e.g., HCOME)?
- b) Also, setion 4.1 is a long section presenting 'validity', but not justifying its importance.
- The title has been changed to "Design …," and an explanation about the method used (phases) to develop our model has been added. This covers the entire process from defining the scope to documentation.
- In subsection 4.1, an explanation of "validity" and its importance for users was introduced. This is particularly crucial for users of road networks who need to know, for example, whether a construction, congestion, or accident is still active, enhancing their ability to make informed decisions.
Pag 5. Line 208. “Detailed documentation of the proposed Linked Open Traffic Data Model can be found”.
In the documentation provided online (with Widoco), I cannot find the evaluation report of OOPS! which is usually included in such a process. Why is that? Have you checked the model against OOPS!?
Yes, it was checked during the project but not included in WIDOCO. Now, it was included the information elaborated by WIDOCO (Errors free). However, when we copy the content of the file on OOPS! the results show falls considered by the tool as important (lack of domains or ranges), however they belong to properties whose unique purpose was to organize the model and properties from the imported ontologies, and so they do not depend on our model. It was explained in a new section of the paper: 5.2 RDF validator and OOPS! (OntOlogy Pitfall Scanner!).
Pag. 9. Line 268. “Units of Measure”. Same problem with 4.1 i.e., a long section presenting only units of measurement. Why?
The title has been changed to “Units of Measure. Reusing QUDT”. Also, it has been explained with an initial paragraph where the existence and significance of packages and data types in the DATEX II standard are discussed, highlighting the need to appropriately address this aspect in the model through existing stable vocabularies.
Pag. 11 Line 305. “Units of GeoSPARQL”. Units? is the term proper? can be improved? e.g., reusing GeoSPARQL elements.
It has been changed to “Geographical aspect. Reusing GeoSPARQL” and explained with an initial paragraph because we decided to use it.
Pag. 13 Line 319. “4.4. Other Sources used. “again, the title can be improved (and remove the full-stop at then) e.g. Reusing other vocabularies
We agree. It has been changed to the suggested title “Reusing other vocabularies“.
Pag 13 Lines: 326-330. many problems in this sentence: a) data ontology? not a a valid term, it can be either data or ontology (schema). b) "as support" to what?. you mean "the ontology". please be careful with capitalization, use only where needed.
It was changed and explained better.
Pag 14. Ontology Editor. there is no need to provide such a long description of the tool used to develop the ontology. After two decades is now well-known and widely-used, especially from the OE community. Also, a link/reference to its web page and related wiki/publications would be sufficient. Figure 11 why they are important for your paper?
We agree. The content related to Protégé was deleted and it was replaced by a new subsection. 5.1 “Conversion processes and validation” has been added, providing a brief overview with some screenshots of the process and technologies used. It does not delve into extensive details as the primary purpose of the paper is to showcase the development process of the ontological model. Also, it was added the subsection 4.2 related to the validation. Figure 11 about statistics and metrics in Portege was replaced by other pictures related to the conversion and validation processes.
Pag 15. Figure 12 and 13. Image is two small for reader to have a look, you need to provide a link with a larger representation of the figure or split the figure in two figures. Also, the link to reproduce the query/result must be available. Same as Figure 12.
The figures were enhanced to address readability issues.
Pag. 16. Figure 14. what is needed to replicate this view in terms of your vocabulary and data? pls state this in the paper clearly.
It was added a brief explanation, and the caption of the figure has been changed.
Pag 16. Line 371. “satellite ontologies”. 'satellite" ontologies? never heard this term in OE. Pls use standard terminology in OE. Also, why these modules are needed to be defined? why these two ontologies have been imported? What was the need?
It was changed to "secondary ontologies". Furthermore, it has been explained in different sections, for example in the introduction and section 4.
Pag 16. Line 381. ”one of the best repositories”. pls remove "one of the best" unless you can justify why it is such.
I agree. It was removed. Thank you for the suggestion.
Pag 6. Line 385. ok, but... from all those classes/properties defined in your ontology, why you have just described a few (Section 4)? how these were selected and why? Why not presenting the main/key classes of your ontology?
Also, what were the Competency Questions that implied the requirements of your ontology? And how these CQ were satisfied (querying the ontology)?
Why there is a lack of description of the requirements of the ontology?
According to your comments, the content of the section 4 has been extended and has been included some aspects related to competency questions, main class etc. in the section 4.
Pag 17. Line 401. “Both technical and legal issues in national and European context were addressed.” but where in the paper are these presented as such?
They are presented in section 2 and now they are explained better.
Pag 17. Line 409. “Increase the size”. Of what?
The amount of data and, consequently, the storage size needed.
The sentence was changed. Now: “This, however, will considerably increase the amount of data and, consequently, the storage size needed and will need to be managed in future.”
Pag 17. Lines 411-413. “the possibilities of implementation of measuring impact”. Syntax.
It was changed to “On the other hand, the possibilities for implementing impact measurement can be summarized in two approaches: measuring the impact of each dataset by making them visible (likes, downloads etc.) or measuring usage through connections to a given system.”

Reviewer 2 Report
Comments and Suggestions for Authors
This paper presents a semantic modelling approach in the transport and traffic sector offering access basic services access to open traffic data. This results in a LOD-enhanced Traffic Information System that complies with the European ITS Directive.
The authors describe how, starting from DATEX II, the standard information model for road traffic and travel information in Europe, they have developed LOD DATEX II as a complementary approach to DATEX II XML. LOD DATEX II relies on a ontological model called dtx_srti that is presented in the paper.
In the introduction, the authors set up the context of the work (Intelligent Transport Systems), introducing the DATEX II XML/UML model that provides a description of concepts and data structures pertaining to traffic. They highlight here the lack of semantics and motivate the choice of the LOD and Semantic Web languages et technologies that will help to solve existing interoperability issues. What is reported in this paper is work accomplished by the authors in the CEF Action LOD RoadTran 18 started in 2018.
In particular, the paper illustrates the development of the semantic model based on the dtax_srti ontology and secondary ontologies that facilitates the mapping between LOD formats and the Safety Road Traffic Information (SRTI) DATEX II profile.
The authors summarise the three main contributions of the paper and an outline of the remainder the paper ends up the introduction.
Section 2 deals with the exploration of legal and technical aspects and directives linked to Public Sector Information and Intelligent Transport Systems. This section is very short and difficult to understand. A list of specifications is provided but the link between these specifications and the model presented in the paper is hard to establish… and not explicitly described.
Section 3 presents semantics issues that were faced. The authors describe how the have to conform to the NTI-RISP standard in Spain for the selection, identification, description, format, use, and provision of public sector documents and information resources. NTI-RISP standard imposes some controlled vocabularies and ontologies that have been adopted by the authors.
A natural question that comes is what if this work have been developed by authors from another European country or if other standard vocabularies or/and ontologies have been imposed? Isn't this the source of a potential interoperability problem, requiring alignment work at global level?
Section 4 is the main section of the article. It presents the Linked Open Traffic Data Model. The designed ontology encompasses the SRTI DATEX II profile and extends it with new concepts and relations. The representation of various traffic incidents is handled by the SituationRecord concept which is central in the dtx_srti ontology that also provides the geographical aspects associated with this traffic incidents. Subsection 1 describes how the W3C Time ontology has been used (although the notion of validity of a situation record is not clearly defined) to integrate temporal components (and their potential relationships as defined by James Allen) that can be queried using SPARQL and the XMLSchema date data type.
In subsection 2, the authors describe how by using the QUDT vocabulary, the ontology associates units of measure with each (quantity) value.
In subsection 3, the authors describe how GeoSPARQL classes have been imported to handle and query geospatial data. Finally, in subsection 4, a list of other vocabularies also used is presented.
Section 5 reports about the validation process the authors have developed to evaluate the dtx_srti ontology. The Protégé editor has been used to build the ontology and to provide some statistics on metrics. In addition, a SPARQL interface has been designed to formulate SPARQL queries and federated queries (using the IGN Spanish national access endpoint) on the data of traffic related to incidents, administrative units and roads.
The LOD View framework was used to generate HTML representations of RDF data (as shown by Figure 14).
Finally, the authors provide the URLs of the ontologies they have developed.
In the conclusion, the advantages and benefits of the LOD approach are outlined. The authors recall that through this project both technical and legal issues in national and European context were addressed. Perspectives concern the measure of the impact of the LOD-SRTI resource and the behaviour of its users.
The paper is well written. Some figures need to be enlarged to get readable without zooming.
Sections 2 and 3 could be summarised and their content includes in the description of the methodological approach.
Although, the technical aspects presented in the paper are not really new nor original the global approach presented here is interesting.
To make the paper more convincing the authors are invited to describe and illustrate how the raw ITS and traffic data (existing or being produced) is transformed into RDF data to populate the model, particularly at the level of the SituationRecord concept.
The English seems to me correct
Author Response
Reviewer #2
Section 2 deals with the exploration of legal and technical aspects and directives linked to Public Sector Information and Intelligent Transport Systems. This section is very short and difficult to understand. A list of specifications is provided but the link between these specifications and the model presented in the paper is hard to establish… and not explicitly described.
In section 2 we expanded the information within each of the subpoints, providing specific details about the analyses considered during the model development. For instance, we detailed the recommendations outlined in Commission Notice 2014/C 240/01, titled "Guidelines on recommended standard licenses, datasets, and charging for the reuse of documents." Additionally, we incorporated specifications related to "Data and procedures for the provision of free Safety-Related minimum Traffic Information (SRTI)" from the delegated act of the ITS Directive. Both were taken into account in the project.
A natural question that comes is what if this work have been developed by authors from another European country or if other standard vocabularies or/and ontologies have been imposed? Isn't this the source of a potential interoperability problem, requiring alignment work at global level?
Thank you for your valuable comment. In the introduction section, we have incorporated details highlighting the noteworthy collaboration between Czech and Spanish participants in working with secondary ontologies for roads and administrative units.
Subsection 4.1 describes how the W3C Time ontology has been used (although the notion of validity of a situation record is not clearly defined).
In subsection 4.1, an explanation about "validity" and its importance for users was introduced. This is particularly crucial for users of road networks who need to know, for example, whether a construction, congestion, or accident is still active, enhancing their ability to make informed decisions.
The paper is well written. Some figures need to be enlarged to get readable without zooming.
Thank you for your valuable comment. The figures were enhanced to address readability issues.
Sections 2 and 3 could be summarised and their content includes in the description of the methodological approach.
The two sections have been retained for clarity, but a paragraph has been added to Section 2 to establish a proper link with Section 3. Additionally, the title of Section 3 has been modified to better reflect its content related to the controlled vocabularies.
To make the paper more convincing the authors are invited to describe and illustrate how the raw ITS and traffic data (existing or being produced) is transformed into RDF data to populate the model, particularly at the level of the SituationRecord concept.
A new subsection 5.1 has been added, providing a brief overview with some screenshots of the process and technologies used. It does not delve into extensive details as the primary purpose of the paper is to showcase the development process of the ontological model.

Round 2
Reviewer 1 Report
Comments and Suggestions for Authors
Thank you for providing the revised version of your paper according to my comments. Please take care of images (still there are few that are too small to read e.g., fig. 1), and also some issues with titles (e.g., too long or with a full-stop at the end).
Comments on the Quality of English LanguagePlease have another round of editing and try to fix a few remaining problems with English.
Author Response
Comments:
Thank you for providing the revised version of your paper according to my comments. Please take care of images (still there are few that are too small to read e.g., fig. 1), and also some issues with titles (e.g., too long or with a full-stop at the end).
Comments on the Quality of English Language. Please have another round of editing and try to fix a few remaining problems with English
Response:
As suggested, figure 1 was enlarged and some headings such as sections 2 and 4 were shortened and their full stop removed. Furthermore, English has been reviewed by our proofreader team, and the proposed improvements have been included.
